# A Review of the Clinical Utilization of Oral Antibacterial Therapy in the Treatment of Bone Infections in Adults

**DOI:** 10.3390/antibiotics13010004

**Published:** 2023-12-19

**Authors:** Nicholas Haddad, Jibran Ajaz, Lina Mansour, Robert Kasemodel, Jennifer Jarvis, John Jarad, Haley Gorski, Maddie Carr

**Affiliations:** 1College of Medicine, Central Michigan University, 1632 Stone St., Saginaw, MI 48603, USA; 2CMU Medical Education Partners, Internal Medicine Residency Program, Central Michigan University, 1000 Houghton Ave., Saginaw, MI 48602, USA; ajaz1sj@cmich.edu (J.A.); kasem1r@cmich.edu (R.K.); jarad1j@cmich.edu (J.J.); 3Covenant HealthCare, 1447 N. Harrison St., Saginaw, MI 48602, USA; lmansour@chs-mi.com (L.M.); madison.carr@chs-mi.com (M.C.); 4Ascension St. Mary’s Hospital, 800 S. Washington Ave., Saginaw, MI 48601, USA; jennifer.jarvis2@ascension.org; 5McLaren Bay Region, 1900 Columbus Ave., Bay City, MI 48708, USA; haley.gorsky@mclaren.org

**Keywords:** oral antibacterial agents, oral antibacterial therapy, oral suppressive therapy, oral antimicrobial therapy, step-down therapy, oral administration, osteomyelitis, bone infection, biofilm

## Abstract

Chronic osteomyelitis in adults is managed with prolonged courses of intravenous antibiotics in conjunction with surgical debridement of necrotic bone. Over the past 40 years, there has been no paradigm shift in this approach, as randomized controlled trials of this standard of care compared to alternatives such as prolonged oral antibiotics are scarce. However, there have been many small trials, case reports, and review papers evaluating the effectiveness of oral treatment for chronic osteomyelitis. The oral route for infections requiring prolonged treatment is intuitively and practically more favorable due to several advantages, the most important of which is the avoidance of long-term IV antimicrobial therapy with its complications, inconvenience, and cost. In this paper, we review the literature evaluating oral antibiotic therapy in the management of chronic bone infections since 1975. The majority of osteomyelitis infections are caused by *Staphylococcus aureus*, hence we focus on its treatment using oral antibiotics; however, we also emphasize subpopulations of patients with diabetes, implanted hardware, and with less common bacterial organisms. The primary objective of this review is to promulgate clinical recommendations on the use of oral antibiotics in bone infections in the context of initial therapy, transition from intravenous therapy, and the role of chronic suppression. The secondary objective is to summarize current knowledge of the specific oral antimicrobial agents that are commonly utilized, together with a synopsis of the available literature pertaining to their pharmacokinetic/pharmacodynamic properties and duration of therapy in bone infection.

## 1. Introduction

The standard treatment for osteomyelitis in adults is prolonged intravenous courses of antimicrobial therapy, typically for four to six weeks [1,2,3,4]. This is based on the belief in the inherent superiority of parenteral antibiotics and concerns about bioavailability and adequate penetration into bone with oral therapy. However, retrospective studies [5,6], clinical trials [7], and case reports [8] have demonstrated that certain oral antibiotics can be utilized as alternatives to IV treatment. These have been shown to be efficacious even in blood stream infections [6,9], bringing oral therapies to the forefront of research and clinical utilization. Notable benefits of the oral route include the convenience, reduced cost, shorter hospital stays, and elimination of the risk of IV-line-associated infections. The high efficacy and minimal toxicity of currently available oral antibiotics are critical factors that have shifted the scales to the enhanced utilization of oral treatment for osteomyelitis. The selection of oral agents should be guided by bioavailability, pharmacokinetic ability to achieve adequate antibiotic concentrations at the site of infection, culture results or local antimicrobial susceptibility profiles if cultures are not available, previous antibiotic response, infection severity, and patient co-morbidities (such as renal function and allergies). Additional benefits include economic advantages [10,11,12,13], improved compliance, comparable efficacy [14,15], and favorable side effect profile [6].

In addition to antibiotics, surgical debridement is essential in removing infected tissue, which prevents cure and may serve as a nidus for recurrent infection. There remain limited data regarding the optimal timing of transition from IV to oral therapy and the duration of oral courses. Osteomyelitis has multiple clinical variations, can affect different bones in the body, and can be caused by many different pathogens. It targets vulnerable hosts such as those with diabetes, neuropathy, trauma, surgery, and spinal cord injury. This variability complicates standardized recommendations for its management, particularly when it comes to the use of oral therapies. The antimicrobials used for bone and joint infections include cotrimoxazole [11,14,16,17], linezolid [16], rifampin [16], fluroquinolones [11,14,17], clindamycin [17], doxycycline [14], amoxicillin [17], amoxicillin/clavulanic acid [17], penicillin [17], and metronidazole [18,19]. The commonly recommended dosing regimens are listed in Table 1.

The optimal time to initiate therapy is after bone cultures are taken; this helps with identifying the causative organism, as this microbiologic knowledge is the cornerstone in treating chronic osteomyelitis. In almost all cases, the antibiotic therapy of chronic osteomyelitis is not an emergency, as the indolent nature of the infection permits waiting for culture results in order to initiate culture-based therapy. The exact duration of antibacterial therapy is not well defined. It is difficult to precisely determine when the bone infection has been eradicated, since the swelling of soft tissue may persist for a long time, reconstruction of cortical bone is often delayed, and laboratory markers (C-reactive protein and erythrocyte sedimentation rate) are of no great value. Treatment can be discontinued upon the resolution of inflammation, evidence of radiological healing, and the absence of clinical relapse, even if the wound has not completely healed. Published evidence suggests that antimicrobial therapy is not to be used for prophylaxis or for healing wounds, but for the treatment of infection [41]. In general, mild infections, in which there is no bone involvement, can be treated with one to two weeks of antibacterial therapy [42,43], whereas two weeks or longer may be required for more severe infections [44]. In the absence of amputation or surgical resection, 6–12 weeks are recommended [45,46,47]. In reviews of postoperative implant infections, the duration of antimicrobial therapy is shorter when implants are removed, such that two to six weeks of IV therapy followed by four to six weeks of oral therapy is recommended. When implants are retained, the initial intravenous therapy is followed by at least 10 weeks, and sometimes lifelong oral therapy [18,21,24,48].

The transition from IV to oral therapy is precluded by the presence of multi-drug-resistant organisms (MDRO) with no oral treatment options, negative cultures (hence no clear microbiologic guidance), allergies to oral options, and adherence concerns [49]. Additionally, while circumventing the complications, expenses, and inconveniences associated with parenteral therapy, unnecessary or inappropriate extended oral antibiotic courses may play a role in the emergence of resistant strains, contributing to the global health concern of antimicrobial resistance. Prolonged antibiotic use also adversely affects the patient with complications such as *Clostridioides difficile* infection and poses other substantial risks that result in higher cost.

This paper reviews publications related to the oral antibiotic treatment of bone infection between January of 1975 and February of 2023. It focuses on osteomyelitis management from multiple perspectives: that of the most common organisms, such as staphylococci and pseudomonas, as well as the less common, such as *Salmonella* and *Kingella* sp. We also focus on populations such as diabetic patients who are particularly prone to bone infection. 

## 2. Results

### 2.1. Efficacy of Oral Antibiotics in Treatment of Osteomyelitis

Oral antibiotics alone were shown in some trials to cure the majority of diabetic foot infections [42,50]. A 2001 report by Senneville et al. on oral rifampin plus ofloxacin for a median duration of six months achieved an 88.2% cure, defined by the disappearance of all signs and symptoms of infection and the absence of relapse during follow up [51]. In a composite review by Conterno et al. (2013), there was no significant difference either in the rates of remission after treatment with oral versus IV antibiotics in chronic osteomyelitis caused by sensitive pathogens, or in the rates of mild, moderate, or severe adverse events between the two groups [7]. Similarly, a 2012 systematic review of studies on osteomyelitis from 1970 to 2011 revealed that the success rates were similar for both routes of administration [1]. 

A multicenter, open-label, parallel-group, randomized, controlled noninferiority trial, Oral versus IV Antibiotics for Bone and Joint Infections (OVIVA) [15], evaluated outcomes at one year of IV versus oral antibiotics for the first six weeks of treatment in 1054 patients. This trial challenged the widely accepted standard of care and concluded that oral antibiotic therapy was noninferior to IV therapy when used during the first six weeks for complex orthopedic infection. Complications were more common in the IV group who also had a considerably longer median hospital stay with no significant difference in the incidence of *Clostridioides difficile*-associated diarrhea, the percentage of participants reporting at least one serious adverse event, or the treatment failure at one year (13.2% in the oral group vs. 14.6% in the IV group). Since its publication, the OVIVA protocol has been implemented in some clinical practices. A study from a British orthopedic hospital showed that patients who were switched to an oral regimen post OVIVA-implementation had a shorter length of stay and reduced cost of care, without a significant difference in clinical outcomes [49]. Highly bioavailable agents with good bone penetration and biofilm activity, such as rifampin and ciprofloxacin, were more commonly used in the oral subgroups. Definitive treatment failure was more common following implementation (13.6% vs 18.6%), although there was no statistical difference between the infection-free survival curves at 12 months, which was similar to those seen in the OVIVA trial [49]. Another study from the Veterans Administration Medical Center in Iowa City implemented a quality improvement protocol aimed at decreasing outpatient parenteral antimicrobial therapy (OPAT) and increasing oral antibiotic use. The outcomes demonstrated significantly lower lengths of stay and no difference in the 6-month recurrence rates or mortality [52]. However, aside from OVIVA, no other large randomized controlled trials to date have determined whether enteral antibiotic therapies are non-inferior to parenteral antibiotic therapies for the empiric treatment of osteomyelitis. Other smaller studies demonstrated the non-inferiority of enteral compared to parenteral therapy for bone and joint infections. Using oral therapies as opposed to IV has significant improvement potential, including reduced utilization of IV lines [53], shorter length of stay, lower cost [49,54], and decreased adverse effect profiles, including nephrotoxicity [55].

In a systemic review and meta-analysis of 25 prospective trials comparing IV-only versus IV antibiotics followed by a stepdown to oral therapy for blood and bone infections, Wald-Dicker et al. found no difference in clinical efficacy between the two groups. In none of the 25 studies was IV-only treatment superior in efficacy [6]. Melis et al. compared oral versus IV antibiotics for hand osteomyelitis in a retrospective study and found no differences in cure rates in patients when followed for at least one year. They defined “cure” by the absence of reactivation or persistence of infection at one year without any need for amputation [14]. For osteomyelitis of the jaw, Lim et al. reported clinical resolution (defined by absence of clinical symptoms and radiographic improvement) in more than 80% of the patients two months after treatment completion when oral antibiotics were used with surgery without any preceding IV antibiotics [17]. Cordero-Ampuero et al followed 36 patients for a minimum of one year to assess if two-stage revision with interim oral antibiotics could eradicate hip arthroplasty infections [28]. Oral antibiotics utilized included ciprofloxacin, rifampin, fosfomycin, doxycycline, clindamycin, trimethoprim-sulfamethoxazole, levofloxacin, and ofloxacin. Eradication was assumed in the absence of clinical, serologic, and radiographic signs of infection. Of the 24 hip arthroplasty infections caused by highly resistant bacteria, 21 of 24 were eradicated using oral antibiotics plus at least a first stage revision (all 12 hip arthroplasty infections caused by sensitive bacteria were eradicated). When considering the 13 hip arthroplasty infections with polymicrobial isolates from intraoperative cultures, 11 were eradicated via oral antibiotics plus at least a first-stage surgery [28]. This demonstrates a high cure rate with oral antibiotics following initial stage revision, even in cases of resistant organisms.

### 2.2. Treatment of Diabetic Osteomyelitis

Diabetes mellitus, according to the National Institute of Diabetes and Digestive and Kidney Diseases, affects 37.3 million people in the United States—approximately 11.3% of the population in 2019 [56]. The most common indications for hospital admission in diabetics are soft tissue and bone infection of the lower limbs [57]. About one-third of diabetics with foot infections have osteomyelitis [42], the most common cause for amputation in the infected diabetic foot. According to the 2020 National Diabetes Statistics Report, 130,000 adults had lower extremity amputations relating to diabetes (5.6 per 1000 adults with diabetes) [58]. Following an amputation, about one-third of patients will undergo an amputation of their other limb within three years, and two-thirds will die in five years [59]. The prompt diagnosis and appropriate treatment of diabetic foot osteomyelitis may help prevent amputation, with its psychological, social, and financial consequences, reduce morbidity and mortality, as well as decrease the burden on the healthcare system.

Emerging evidence reveals that most infections respond well to antibiotics alone [60]. The early excision of all infected bone is deemed necessary by some authors, only necrotic bone removal is suggested by others, and limited debridement in the clinic—with surgery restricted to patients who are unresponsive to antibiotics—is suggested by others still [61]. The conventional advice that that excision of infected and necrotic bone along with aggressive parenteral therapy is crucial in the management of diabetic foot osteomyelitis was challenged by a 10-year retrospective review [46]. The majority of the 22 patients identified to have overt osteomyelitis were successfully managed with prolonged courses (12 weeks median) of oral antibiotics (most commonly, clindamycin 600–1800 mg daily) as outpatients with limited debridement undertaken in the clinic, without the need for hospitalizations for either debridement or parenteral antibiotics. Data from another study [61] demonstrated similar results, suggesting that conservative management with antibiotics alone, whether oral or IV, can be successful in the majority of cases, except when surgery is clearly indicated. An analysis published in JAMA in 1995 concluded that a 10-week culture-guided post-surgical debridement oral antibiotic therapy, in patients without systemic infection, may be as effective and less costly than other approaches [62]. A retrospective medical record review of 325 diabetic patients with foot osteomyelitis revealed successful treatment with oral antibiotic therapy, with or without debridement, in almost 80% of cases, with a mean duration of therapy of 40 ± 30 weeks. This study also concluded that acceptable results from oral therapy may be attained even with little operative facilities and resources [63].

While mild and non-limb-threatening infections are generally monomicrobial and can thus be treated successfully with a single antibiotic, severe and/or limb-threatening infections are usually polymicrobial, involving both aerobic and anaerobic organisms [45,64,65,66]. Gram-positive cocci including *Staphylococcus aureus*, coagulase-negative staphylococci, group B streptococci, *Enterococci*, and *Corynebacterium species* are the most common cultured organisms, but gram-negative bacilli and/or anaerobes may frequently be encountered as well. Accordingly, empiric coverage targeting gram positives and gram negatives, as well as aerobes and anaerobes is recommended in most situations. A prospective, randomized, multicenter trial comparing the efficacy of two broad-spectrum regimens initially administered intravenously (ofloxacin vs ampicillin/sulbactam) then orally (ofloxacin vs amoxicillin/clavulanate) to two treatment groups with similar infection severity showed that 67% of the pathogens were gram-positive cocci and 92% were aerobic organisms. Both regimens proved to be effective after a total duration of about three weeks [67]. In some settings, however, empiric therapy needs to be different from the published guidelines, depending on the local prevalence of microorganisms. The results of a multicenter descriptive and analytic cross-sectional study from 17 centers of four Latin American countries found that most infections, unlike in other continents, were monomicrobial, and gram negatives had a high prevalence in mild diabetic foot infections. A combination empiric treatment, amoxicillin-clavulanate with trimethoprim-sulfamethoxazole, was suggested for diabetic foot infection if osteomyelitis is probable [68]. In Europe, rifampin is used as adjunctive therapy to the backbone oral antimicrobial treatment for osteomyelitis, including diabetic foot, and the results suggest that it may improve amputation-free survival [69]. While mild and moderate infections may be treated with oral antibiotics alone, oral therapy may not be appropriate for patients with systemic illness, severe infection, poor enteral absorption, vasculopathy, or infections caused by organisms resistant to oral agents. Additionally, with the prevalence of comorbid peripheral artery disease in diabetic foot infections, diabetic patients with foot osteomyelitis should be evaluated for limb ischemia and undergo revascularization as appropriate [70]. With concomitant revascularization, patients may be successfully treated with medical therapy and avoid amputation [8]. Until proper blood supply is established, diabetes is a risk factor for treatment failure in chronic osteomyelitis treated with prolonged suppressive oral antibiotics [3]. A few case reports describe the successful treatment of osteomyelitis of the proximal and distal phalanx of the toes with IV antibiotics followed by 10 weeks of oral antibiotics after successful angioplasty. Optimal outcomes in those reports are defined as resolution of the ulcers, radiographic defects, and complete restoration of foot function [8]. The paucity of similar case reports renders such an observation less robustly generalizable. The treatment of osteomyelitis using orally administered antibiotics is also favorable in the high-risk niche persons who inject drugs (PWID). A retrospective analysis by Marks et al. showed that in PWID with invasive bacterial infections who left against medical advice, prescribing no oral antibiotics at discharge compared to oral antibiotics, was associated with 2.32 times higher odds of 90-day readmission. They also noted that the 90-day admission rate was similar in people who were discharged on oral antibiotics compared to those had a full IV antibiotic course [71], another indirect but strong indication that IV and oral therapies are equally efficacious in this subpopulation of patients. 

Oral antibiotics for the treatment of osteomyelitis in specific bones have also been described, primarily through case studies and smaller series. For example, in jawbone and joint infections, common pathogens include viridans *Streptococci*, mixed anaerobic flora, coagulase-negative *Staphylococcus*, *Actinomyces*, *Eikenella corrodens*, *Candida species*, *Neisseria species*, *Enterobacteriaceae*, and *Staphylococcus aureus* [17]. The retrospective analysis of jaw osteomyelitis by Lim et al. showed better outcomes at two months posttreatment in patients treated with oral antibiotics after surgery compared to those treated with IV antibiotics. The oral antibiotics that have been successfully used for jaw osteomyelitis treatment include amoxicillin, amoxicillin/clavulanic acid, clindamycin, moxifloxacin, cotrimoxazole, and penicillin [17]. As for bone and joint infections in the hand, a retrospective study of 61 patients with acute inoculation osteomyelitis of the hand were treated with oral antibiotics for six months; this resulted in a 100% cure rate [11]. The antibiotics used included trimethoprim/sulfamethoxazole, levofloxacin, and moxifloxacin. Common pathogens were methicillin-resistant *Staphylococcus aureus* (MRSA) (22%), *Staphylococcus epidermidis* (18%), methicillin-resistant *Staphylococcus aureus* (MSSA) (13%), and *Streptococcus species* (10%).

### 2.3. Treatment of Staphylococcus aureus as Leading Causative Organism

*Staphylococcus aureus* is a leading cause of bone and joint infections (BJI), culture proven in as many as 75% of cases. *Staphylococcus species* have evolved over the decades to acquire resistance to beta-lactam antibiotics, such that *Staphylococcus aureus* is distinguished as being methicillin sensitive or methicillin resistant. More than one-third of all staphylococcal BJIs are caused by MRSA, with an increasing prevalence of methicillin resistance [72]. Physicians have historically treated BJI with empiric parenteral antibiotics that are effective against MRSA, and de-escalated only if MRSA is ruled out based on culture and/or molecular testing data. This practice came about when many of the currently available anti-MRSA enteral antibiotics did not yet exist. The recent increasing availability of efficacious and highly bioavailable oral antibiotics in the management of MRSA infections has challenged this historical practice [73]. One important gap in the knowledge is whether multi-drug enteral regimens are superior to monotherapies because it has long been known that causative organisms of BJIs create biofilms that act as a mechanical barrier and prevent the delivery of the antibiotic to the nidus of infection. Some antibiotics, like rifampin, penetrate these biofilms and work synergistically with other agents to overcome this bacterial defense mechanism [74].

For the initial treatment of chronic osteomyelitis caused by MSSA, parenteral beta-lactam agents, i.e., oxacillin, nafcillin, and cefazolin, are suitable options. Unfortunately, nafcillin and cefazolin are not available in an oral formulation, and the bioavailability of oral penicillin, oral formulations of oxacillin, and cephalosporins, is usually low. Therefore, switching to oral therapy often requires a change to other agents active against both MSSA and MRSA, such as doxycycline, clindamycin, linezolid, and/or trimethoprim-sulfamethoxazole. Rifampin should be combined with other agents, namely fluoroquinolones or linezolid [2]. Hence, rifampin has a niche as a “biofilm active agent”. It is best studied for staphylococcal prosthetic joint infection in the setting of hardware retention [75,76,77]. Data were accordingly extrapolated for other hardware infections such as osteofixation and spinal implant. Its use results in lower treatment failures in prosthetic joint infections (PJIs) with implant retention. However, it has significant pharmacologic interactions (primarily via the CYP 3A4 pathway) that need to be considered prior to its clinical utilization. Published studies recommend against rifampin use in combination with fusidic acid [78,79], the use of which is uncommon in the United States. Another caution is co-prescribing oral clindamycin and rifampin, as clindamycin concentrations can be substantially decreased due to increased first pass metabolism resulting from P450 enzyme induction by rifampin [80]. However, outside of those cautions, the use of rifampin is an important adjunct in the management of staphylococcal osteomyelitis, particularly those involving biofilms, such as with prosthetic implants. Discouraged rifampin usage includes monotherapy (due to prompt risk of resistance emergence) [81] and prior to surgical debridement without a partnered antimicrobial. When choosing between oral antibiotics, a retrospective analysis by Nguyen et al. found the rifampin-trimethoprim-sulfamethoxazole combination to be as effective as rifampin-linezolid, with trimethoprim-sulfamethoxazole being cheaper than linezolid [16].

In a study from France, levofloxacin was used in combination with rifampin for susceptible staphylococci. The dose of levofloxacin was increased when patient weight exceeded 100 kg (to 750 mg) or decreased if creatinine clearance was less than 30 mL/min (to 250 mg). Only one of the 79 treated patients failed treatment, this may have been due to implant retention [37]. However, in other studies, failures were ascribed to treatment of staphylococcal osteomyelitis with ciprofloxacin monotherapy. Dellamonica and colleagues assessed 39 patients with chronic osteomyelitis that had persisted for at least two months. A total of 7 of the 39 were treated with ciprofloxacin, and 3 of those failed, as ciprofloxacin in those cases was used as monotherapy for *Staphylococcus aureus* [34]. In another study, 30 adults were randomized to receive oral ciprofloxacin (750 mg every 12 h), or other antimicrobial therapies, for the treatment of osteomyelitis. The duration of treatment lasted anywhere between 19 and 150 days. Cure was achieved in only 50% (7/14) of the ciprofloxacin-treated patients and in 69% (11/16) for other antimicrobial therapies. However, the cipro treatment failed in one *S. aureus* infection [35]. In a study involving 61 patients with primary vertebral osteomyelitis, Babouee et al. assessed switching IV to oral antibiotics. Most patients’ antibiotic therapy (72%) was switched to oral after four days of IV therapy. In 21 patients, the switch to oral therapy occurred after two weeks. The most frequently used oral therapy was ciprofloxacin with or without rifampin. The average duration of antibiotic therapy was 57 days. *Staphylococcus aureus* (21%) and coagulase-negative staphylococci (17%) were the most frequently isolated microorganisms, followed by gram-negative bacteria (28%), streptococci (20%) and *Propionibacterium acnes* (5%). In eight patients, no microorganisms could be identified. This study showed that switching to an oral antibiotic regimen after two weeks of IV therapy may be safe if symptoms have improved, the epidural or paravertebral abscess has been drained, and C-reactive protein levels have decreased [26].

Charalambous et al. conducted a single-institution retrospective cohort study between 2015 and 2020 for patients who developed a prosthetic joint infection (PJI) after primary or previous revision total knee replacement with monomicrobial isolates of coagulase-negative staphylococci (*Staphylococcus epidermidis*, *Staphylococcus lugdunensis*, *Staphylococcus schleiferi*, *Staphylococcus caprae*, *Staphylococcus mitis*, *Staphylococcus hominis*) [82]. Patients with a history of prior PJI in the same joint were less likely to be clear of infection at one year postoperatively. Vancomycin was the most prescribed IV antibiotic, followed by cefazolin and daptomycin. Out of the 55 patients included in the study, there were a variety of antibiotic modalities utilized: 21 patients only completed IV therapy, 13 patients required lifelong chronic oral antibiotic suppression, 20 patients received some combination of IV and PO antibiotics, and only one patient received oral antibiotics only. Eight patients received adjunct rifampin for an average duration of eight weeks. Less than 50% of patients achieved one-year infection clearance; this was defined as not undergoing revision surgery, not growing positive microbiological cultures indicative of persistent coagulase-negative staphylococcal infection, and not having recurrent PJIs, and not needing chronic suppressive oral antibiotic regimen in the year after their PJI initial treatment. They found that the two-stage revision had better overall one-year infection clearance than debridement, antibiotics, and implant retention [82].

### 2.4. Treatment of Pseudomonas aeruginosa and Other Gram-Negative Etiologies of Osteomyelitis

Given their favorable PK/PD data, fluoroquinolones have been well studied and have shown great efficacy in the management of osteomyelitis. They are bactericidal against most gram-negative aerobic bacilli, with ciprofloxacin having specific indications for the treatment of *Pseudomonas aeruginosa*. Previous literature has shown that quinolones are successful in treating pseudomonal and other gram negatives, in addition to other organisms, such as *Staphylococcus aureus* [83,84,85]. Papers published in the early 1990s highly recommended quinolones. They demonstrated success for the treatment of pseudomonal and other gram negatives, in addition to other organisms such as *Staphylococcus aureus*. Most of those references touted the favorable adverse effect profile of fluoroquinolones, with almost no reports of tendon damage at that time [86,87,88]. Those recommendations have changed over time—now fluoroquinolones are generally considered effective second line agents for infections caused by sensitive organisms [89]. Some of those early papers cautioned that the indiscriminate use of fluoroquinolones carries inherent concerns of potential failure in the treatment of staphylococcal osteomyelitis [40], over time falling completely out of favor for the treatment of staphylococcal infections. Of the quinolones, ciprofloxacin has been studied the most [83,86,90]. In 1988, the Swedish Study Group reviewed the use of oral ciprofloxacin in the management of gram-negative osteomyelitis [83]. This was an open, non-comparative, multi-center trial in 17 Swedish hospitals that evaluated a total of 34 patients with osteomyelitis in different bones. Patients were treated with various doses of ciprofloxacin without any parenteral therapy. There was resolution in 22 (65%) patients, improvement in 5 (14%), and failure in 7 (21%); thus ciprofloxacin was recommended as an oral alternative to IV antibiotics for the treatment of acute or chronic osteomyelitis caused by sensitive gram-negative bacilli including *Pseudomonas* sp. [88]. In 1990, Gentry et al. demonstrated the safety and efficacy of oral ciprofloxacin compared to IV therapy for infections caused by a wide variety of organisms, with a high success rate of 77% and no significant adverse events [86].

Oral ciprofloxacin in doses of 750 mg twice daily for the treatment of chronic osteomyelitis was assessed in hospitalized adult patients who presented with chronic osteomyelitis in bone biopsy and an organism susceptible to ciprofloxacin. When *Enterococcus faecalis* was isolated, ampicillin was added after ensuring that it had no activity similar to the test drug against the Gram-negative organism. The duration of treatment ranged from 28 to 254 days. The oral treatment with ciprofloxacin proved to be useful for the prolonged therapy of chronic osteomyelitis, always combined with surgical debridement. Oral therapy allows for easy outpatient use, good tolerability, and is effective; however, it demands special attention for the possible emergence of resistance, particularly in *Staphylococcus aureus*. It is important to note that 11 patients (65%) had been treated with other antibiotics before admission to the study [30]. 

A group of 14 patients with chronic osteomyelitis were treated with oral ciprofloxacin (group I) and compared with a group of 12 patients of similar age who had chronic osteomyelitis and received standard parenteral antibiotic therapy consisting of nafcillin, clindamycin, and gentamicin, singly or in combination (group II). The osteomyelitis was successfully treated at the end of therapy and upon follow-up after therapy completion in 11 patients in the first group and 10 in the second group. The average duration of antibiotic therapy (38 days) and follow up (approximately 30 months) were about the same for both groups. The oral administration of ciprofloxacin was as effective and safe as parenteral therapy for the treatment of osteomyelitis in these adults [33]. At a large French university hospital, a cohort of 67 patients with *Pseudomonas aeruginosa* osteomyelitis was identified over a 15-year period. All but one patient were treated with a combination of surgery and antibiotic therapy, and they had an overall treatment success rate of 79.1%. The authors propose surgical debridement with IV antibiotics for no more than 15 days, followed by an oral fluoroquinolone with a maximum duration of six weeks [25]. Eron et al. assessed the use of oral ofloxacin for infections caused by bacteria resistant to oral antimicrobial agents, including the most common pathogens *P. aeruginosa*, *S. aureus*, and *E. coli* [39]. Patients with infections caused by *Enterobacteriaceae*, *Staphylococcus aureus*, and *Enterococcus faecalis* were effectively treated. Ofloxacin therapy resulted in unsuccessful results when patients had infections due to *P. aeruginosa*, with resistance emerging usually one month into therapy [39]. Table 2 summarizes published retrospective, prospective, and randomized studies on the treatment of osteomyelitis since 1987.

### 2.5. Treatment of Osteomyelitis Caused by Less Common Organisms

There is paucity of evidence in the literature for the definitive treatment recommendations of unusual organisms. Reliance on the antibiogram and “standard” antimicrobial regimens for those organisms is what guides the choice of treatment. 

Infections caused by *Kingella kingae* typically occur in patients with immune-compromising conditions. This organism is classically associated with endocarditis, bacteremia, and spondylodiscitis in adults. In a case report by Wilmes et al, high-dose oral amoxicillin therapy was used for pubic osteomyelitis, without surgical debridement, with good clinical and radiographic response to treatment after three months [22]. 

Osteomyelitis and septic arthritis caused by *Yersinia enterocolitica* is rare. Its most common portal of entry is the GI tract following the ingestion of contaminated food, water, or milk. Isolates of serotypes O3, O9, and O8 are the most frequent causes of sporadic human disease worldwide. This organism may cause infections in individuals without major underlying disease or specific risk factors. In a study where oral ciprofloxacin therapy was used with therapeutic success. The drug of choice is yet to be identified [29]. 

Discitis and vertebral osteomyelitis caused by *Fusobacterium nucleatum* have been treated with IV ertapenem for eight weeks initially together with oral amoxicillin/clavulanate for a total of 10 weeks. At 1-month follow-up after the completion of treatment, the patient’s inflammatory markers returned to normal values, and the infection resolved with L3–L4 auto-fusion [23].

Group G streptococcal osteomyelitis is rare, with fewer than 15 cases reported in the literature. A case report of a 71-year-old otherwise healthy male with osteomyelitis of the proximal femur was treated with IV penicillin for six weeks, followed by oral cephalexin for six months with a good outcome. The optimal dosage and duration of antibiotic therapy for group G streptococcal osteomyelitis, as well as the role of surgical debridement, are controversial [94]. 

An extremely rare case of *Salmonella potsdam* vertebrae osteomyelitis was confirmed via tissue culture and abscess fluid obtained during surgery. Based on a drug-sensitivity test, levofloxacin and ceftazidime were administrated through IV injection for three weeks, followed by oral antimicrobial therapy for another three weeks. At the 4-month follow up, back pain had almost completely resolved; the patient’s MRI demonstrated an improvement of swelling, with noted radiographic changes of edge sclerosis and L4/5 partial fusion [95].

*Aggregatibacter actinomycetemcomitans* is well known as the pathogen behind gingivitis and periodontitis. Discitis and vertebral osteomyelitis cases caused by this organism have rarely been reported. A successful antimicrobial therapeutic strategy for discitis with this organism is ampicillin or amoxicillin, but no cases have been reported using levofloxacin. Uno et al. reported a case where levofloxacin was selected due to unclear susceptibility results to amoxicillin (the organism failed to grow). It was used after two weeks of IV ceftriaxone. It was originally prescribed for two weeks, yet it was extended to six weeks due to elevated CRP and exacerbated low back pain two weeks post discharge. Based on that report, levofloxacin, to which *A. actinomycetemcomitans* is usually susceptible, can be an effective alternative oral antimicrobial agent when amoxicillin or ampicillin cannot be utilized. Six-week parenteral or highly bioavailable oral treatment is recommended in the case of discitis or vertebral osteomyelitis. This organism is usually susceptible to cephalosporins, rifampin, tetracyclines, or fluoroquinolones, and in vitro susceptibility to penicillin and ampicillin is variable. However, the clinical efficacy of fluoroquinolone therapy for this organism cannot be generalized based on this report, but it should be considered if no other options are available [36].

### 2.6. Antimicrobial Bone Levels

Malincarne et al. in an open three-armed non-randomized trial, evaluated moxifloxacin penetration into bone to evaluate its potential role in the treatment of bone infections. They determined plasma and bone moxifloxacin concentrations following the oral administration of single or double doses (400 mg every 12 h) [96]. The recovered plasma and bone concentrations after single administrations showed a stable bone/plasma ratio without a relevant reduction in plasma or tissue drug levels. Considering an MIC90 of 0.12 mg/L for methicillin-susceptible staphylococci and of less than 1 mg/L for most Enterobacteriaceae, the recovered mean moxifloxacin concentrations show that single dosing leads to bone and plasma moxifloxacin levels exceeding the MICs for the most relevant pathogens. Double moxifloxacin administration gives significantly higher plasma and bone concentrations, with an average of above 2.5 mg/L both in cancellous and cortical bone. This value exceeds the MIC90 of moxifloxacin for methicillin-resistant staphylococci, as reported by most authors, of 2 mg/L, and the clinical *M. tuberculosis* susceptibility breakpoint for moxifloxacin of 1 mg/L. The results demonstrate a good degree of penetration of moxifloxacin into bone [96].

Despite concerns about its bone penetration and poor bioavailability [97], amoxicillin/clavulanic acid, with its beta-lactam/beta-lactamase combination among the most frequently prescribed oral beta-lactam antibiotics worldwide for diabetic foot infections. It is considered the drug of choice by some, with activity against MSSA, streptococci, enterococci, many gram-negative rods, and anaerobes [98,99]. A retrospective cohort analysis among 794 diabetic foot infection episodes—including 339 diabetic foot osteomyelitis cases— found that oral amoxicillin/clavulanic acid resulted in similar clinical outcomes to other regimens and was a reasonable option when treating diabetic foot infections and osteomyelitis [100]. Other antibacterials used orally with successful outcomes are clindamycin (especially in patients with penicillin allergy) and fluoroquinolones (for gram-negative infections or in combination with amoxicillin/clavulanic acid or clindamycin). For limb-threatening infections, clindamycin, an aminoglycoside, and ampicillin as a triple antibacterial regimen have been found to be successful. 

### 2.7. Economic Perspectives

Multiple cost analysis studies have demonstrated the superiority of oral antibiotics for osteomyelitis. In a retrospective study by Bhagat et al. 73% of patients receiving outpatient IV antibiotics (OPAT) were found to be candidates for oral antibiotics per OVIVA criteria, and substituting oral for IV antibiotics could have resulted in an estimated average savings per patient of USD 3270.69 [10]. A similar analysis by Marks et al. conducted in the United Kingdom, found that 79.7% of patients were eligible for oral antibiotics, with an estimated savings of GBP 2950 (USD 3605) [12]. A scenario analysis for early discharge and outpatient oral treatment for osteomyelitis for the National Health Systems (NHSs) in Italy, Greece, and Spain suggested a positive impact in terms of the incidence of hospital-acquired infections, hospital bed saving/increased productivity, and reduced direct health care costs [13]. In their study of patients being treated for osteomyelitis of the hand with oral antibiotics, Henry et al. calculated a differential direct cost savings of 98% when compared to IV therapy [11]. In their systematic analysis of the 25 prospective studies comparing IV-only therapy to IV followed by oral stepdown, Wald-Dicker et al. also reported prolonged inpatient hospitalization in the IV-only group [6], which would translate to increased healthcare utilization costs.

## 3. Discussion

Oral antibiotics have been demonstrated to have comparable efficacy in treating osteomyelitis in different populations (diabetes, traumatic osteomyelitis, and osteomyelitis associated with hardware) and anatomic contexts [6,14,15,17]. The advantages of using oral over IV antibiotics include healthcare cost savings [10,11,12,13], shorter hospital stay [6,13], and the decreased incidence of catheter-related adverse events [12]. This also translates to decreased burden on the hospital systems and increased bed availability, the benefits of which will extend to patients beyond those with osteomyelitis [13]. The use of oral antibiotics is especially important in PWID with bone and joint infections. When they leave the hospital (even if against medical advice) without outpatient IV antibiotic arrangements but with oral antibiotic treatment, their re-hospitalization rates were demonstrated to be significantly reduced [71]. 

Outcomes have been favorable in most reports of oral antibiotics in osteomyelitis. In published studies, the range of cure, defined as complete resolution of symptoms, clinical, and radiologic findings, is between 66 and 100% (66% [4,83], 77% [101], 79% [4], 81% [102], and 90% [92,103,104]). The duration of oral therapeutic regimens in those studies was at least six weeks. With that, more authors conclude that there is no statistically significant difference between parenteral and oral antibiotics for the treatment of osteomyelitis if the microorganisms being treated were sensitive to the antibiotic used and the oral antibiotic has excellent bioavailability [1,7]. Many authors recommend chronic suppressive therapy after initial IV therapy with an oral agent for three to six months, primarily when surgical debridement is suboptimal. This includes when the patient is not willing or able to undergo surgical resection, if removal of the prosthetic material or infected sequestrum is unachievable, or due to poor surgical candidacy [2,6]. 

Ancillary management with surgical debridement is a critical component for success when there is a sequestrum, which is the natural pathophysiology in chronic osteomyelitis. Hence, success is achieved with serial surgical procedures with a potential need for bone grafting and local muscle flaps [105]. 

The use of oral antibiotics is a clinical decision that should be primarily based on the culture results and sensitivity of the organism, and secondarily on a myriad of other factors such as: the bioavailability of the antibiotic of choice, patient compliance, functionality of the gastrointestinal tract, allergy profile, and opinions of specialty consultants. It is not necessary to follow serum bactericidal levels because the failure of therapy is usually due to suboptimal surgical debridement rather than inadequate antimicrobial therapy. Although failure could be attributed to the persistence of the infection in a shielded location –such as biofilms [75,106]—or to the resistance of few bacterial colonies, the efficacy of prolonged oral suppressive therapy is thought to be due to the prolonged action against bacterial replicating at a slow rate, or to action against suspended bacterial cells liberated from the glycocalyx [4].

## 4. Materials and Methods

We reviewed the Ovid MEDLINE(R) and Epub Ahead of Print, In-Process, In-Data-Review & Other Non-Indexed Citations, Daily and Versions starting 1946 through 8 February 2023. Search strategy included all the key terms relating to our search: oral antibacterial agents, oral antibacterial therapy, oral suppressive therapy, oral antimicrobial therapy, step-down therapy, oral administration, osteomyelitis, bone infection, and biofilm. We limited our search to the English literature, and adult patients. We initially had 560 papers. The reason for this was that “oral” ended up pulling a multitude of papers that dealt with oral infections, so upon review for relevance, we were able to eliminate the irrelevant papers, and had 160 that specifically discussed oral antibiotic therapy in adult osteomyelitis. We divided those into categories, and each of the authors reviewed an average of 20 papers. From those, multiple other papers were pulled from the references. We consequently ended up reviewing a total of 106 references for this review.

## 5. Recommendations

Enteral agents (alone or in appropriate combinations) recommended in the management of osteomyelitis include fluroquinolones, cotrimoxazole, clindamycin, doxycycline, amoxicillin, amoxicillin/clavulanic acid, linezolid, rifampin, and metronidazole; these agents have excellent bioavailability, and their use facilitates safe outpatient treatment, avoiding the inherent risk of prolonged IV access.Enteral linezolid, clindamycin, trimethoprim–sulfamethoxazole, and doxycycline are effective treatment options for staphylococcal osteomyelitis when culture and sensitivity data corroborate their use at any stage of infection.Treatment of *Staphylococcus aureus* osteomyelitis with fluoroquinolones is associated with higher failure rates and should be generally avoided as a monotherapy.Rifampin is considered a niche as a biofilm-active agent, best to be considered for staphylococcal prosthetic joint infection in the setting of hardware retention, where its use results in lower treatment failures.Enteral ciprofloxacin is the agent of choice in management of pseudomonal osteomyelitis, when oral therapy is deemed appropriate, and the organism demonstrates sensitivity to quinolones.In the management of osteomyelitis, surgical debridement is a cornerstone of therapy, together with antimicrobial therapy. Most studies reviewed indicate that a few days to few weeks of culture-based intravenous therapy, then transitioning to oral therapy, is effective in achieving long-term cure. Very few studies started with oral antibiotics alone, such that no generalizable recommendation can be made regarding the exclusive utilization of oral antimicrobial therapy, although this could be appropriate in certain clinical circumstances.In patients on IV therapy for chronic osteomyelitis caused by pathogens sensitive to oral antibiotics, consideration should be given to continuation with an oral agent to which the pathogen is confirmed sensitive. This is based on ongoing demonstration in the literature over many years that there is no significant difference in rates of remission after treatment with oral versus IV antibiotics, granted that the oral agent has an established high bioavailability profile and tolerance by the patient. Duration may need to be prolonged; studies quote anywhere between 6 weeks and 6 months, depending on clinical response. This wide range is related to the heterogeneity of patients in those studies.In patients who start IV therapy and are potentially good candidates for enteral therapy, transitioning from IV to oral therapy can be considered without time considerations once full source control (debridement) has been achieved, symptoms have improved, and inflammatory markers have decreased. Many of the published studies suggest a time frame of two weeks as optimal for this switch to occur. Some studies compared earlier transitioning to enteral therapy within the first week without significant clinical difference in outcome. The current thought on this based on the above review is that two weeks of IV therapy may be necessary for most patients before any consideration of switching to oral treatments. There is ample evidence so far to prove similar long-term efficacy in the cohorts switched to oral therapy early during their six-week course as compared to those who continued IV.After initial IV treatment of patients with implant-related infection, oral suppressive antibiotics should be initiated until all implants are removed. In patients where implants cannot be completely removed, lifelong oral suppressive antibiotic therapy should be considered.In diabetic patients, empiric addition of rifampin should be a consideration as adjunctive therapy to backbone culture-based oral antimicrobial treatment, as it may improve amputation-free survival.In diabetics without known vascular disease, culture-based oral antimicrobial therapy can be safely used as an alternative to IV therapy in most situations.Contraindications to the use of oral antibiotics alone include osteomyelitis associated with severe systemic illness, poor enteral absorption, vasculopathy prior to surgical correction, and infections caused by organisms resistant to oral antimicrobials.Populations who may particularly benefit from oral therapy compared to IV are persons who inject drugs, where oral antibiotics prescribed at discharge were associated with significantly less than no antibiotics in regard to 90-day readmission, and that was comparable to continuation of IV treatment after discharge.Mandibular and hand osteomyelitis, typically caused by trauma in otherwise healthy individuals, can be effectively treated with oral regimens, as has been shown by multiple small studies and case series using amoxicillin, amoxicillin/clavulanic acid, clindamycin, moxifloxacin, levofloxacin, trimethoprim-sulfamethoxazole, and even penicillin VK.Recommendations for treatment in specific situations/pathogens:*Aggregatibacter actinomycetemcomitans* discitis: oral levofloxacin for six weeks.*Salmonella* vertebraal osteomyelitis:chloramphenicol, third-generation cephalosporins, and fluoroquinolones for at least six weeks.*Fusobacterium nucleatum* discitis and vertebral osteomyelitis: IV ertapenem for eight weeks in combination with oral amoxicillin/clavulanate as oral suppression for a total of 10 weeks.*Yersinia enterocolitica* septic arthritis and osteomyelitis: oral ciprofloxacin.Group G streptococcus osteomyelitis of the proximal femur: IV penicillin for six weeks and oral cephalexin for another six months.*Kingella kingae* pubic osteomyelitis with soft tissue abscess: high-dose PO amoxicillin 3 g for three months.

## 6. Conclusions

In the majority of the reviewed papers, there were no clinically significant differences between oral and parenteral antibiotics for the treatment of osteomyelitis if the targeted pathogen(s) were sensitive to the antibiotic(s) being utilized. Endpoints for response were defined differently in different papers, but were comparatively achieved (Table 2). These results outline the importance of pathogen-specific therapy, which could be enteral as long as other patient and drug related factors are taken into consideration. When used in the appropriate patient and condition, oral treatment is a welcome alternative to IV therapy, with ample advantages and generally equally favorable outcomes. 

The strength of this review is that it displays the myriad studies which discuss the clinical efficacy and advantageous profiles of oral vs. parenteral antibiotics for treating osteomyelitis in adults—a discussion that ultimately leads to distillation of those papers into clinically applicable recommendations. However, a few questions remain largely unresolved, such as: timing considerations for starting with oral therapy vs. switching to oral at a later stage, the outcomes of patient populations that transition earlier to enteral therapy, the most appropriate length of treatment with oral agents, and the endpoints most relevant for defining a durable cure after prolonged post-therapy follow up. Limitations include the lack of consistency in the published literature and large-scale randomized controlled trials. Accordingly, future clinical research in large cohorts of patients with osteomyelitis comparing IV vs. oral therapies is needed to define the role, efficacy, and timing of oral therapy, Additionally, larger multicenter studies are required to better understand the nuances in antibiotic regimens in varying patient groups and against specific pathogens. This is important to achieve for the ultimate intent of establishing definitive evidence-based treatment guidelines in the management of patients with osteomyelitis using oral antimicrobials. 

## Figures and Tables

**Table 1 antibiotics-13-00004-t001:** Commonly recommended dosing regimens of oral antibiotics in the treatment of osteomyelitis.

Drug (Class)	Oral Dosing for Bone Infection in Normal Kidney Function q = Every, h = hours	Targeted Organism(s)
Amoxicillin [20,21,22]	3 g daily 1 g q8 h	*Kingella kingae* *Streptococcus* *Enterococcus* Gram-positive anaerobes
Augmentin [23]	875/125 mg q12 h Renal adjustment	*Fusobacterium*
Bactrim [24]	1 DS tab q12 h Renal adjustment	*Staph aureus* and/or Enterobacteriaceae (in combination with rifampin for staph infections)
Ciprofloxacin [20,21,25,26,27,28,29,30,31,32,33,34,35]	750 mg q12 h 500 mg q12 h Renal adjustment	Gram-negative bacilli *Staph aureus* (in addition to rifampin) *Staphylococcus epidermidis* *Enterococcus* *Pseudomonas aeruginosa*
Clindamycin [24,28]	300–450 mg q6-8 h	*Cutibacterium acnes* MRSA
Doxycycline [20,21,24,28]	100 mg q12 h 100 mg q8 h No dose adjustment	*Staph aureus* (in combination with rifampin) Gram-positive anaerobes
Levofloxacin [20,21,26,27,28,36,37]	750 mg daily 500 mg daily 500 mg q12 h Renal adjustment	Gram-negative bacilli *Streptococcus* *Aggregatibacter actinomycetemcomitrans*
Linezolid [20,21,28]	600 mg q12 h	*Enterococcus*
Metronidazole [20,21]	500 mg q8 h	Gran-negative anaerobes (*Bacteroides, Fusobacterium*)
Ofloxacin [28,34,38,39,40]	400 mg q12 h 200 mg q12 h 200 mg q8 h	*Pseudomonas aeruginosa* *Staph aureus* *Enterococcus* Gram-negative bacilli
Rifampicin [24,26]	600 mg daily No dose adjustment	Biofilm—used in addition to anti-staph agent
Rifampin [20,21,28]	450 mg q12 h 300 mg q8 h	*Staphylococcus* Gram-positive anaerobes

**Table 2 antibiotics-13-00004-t002:** Summary of studies between 1987 and 2023 of osteomyelitis treatment (oral/IV combinations).

Author and Citation	Number of Patients in Trial	IV Drug(s) and Treatment in Days	PO Drug(s) and Treatment in Days	Severe AE	Outcomes (% Cured)	Study Design
Nix 1987 [88]	37	None	Cipro 139 days	3	31 (84%)	Retrospective
Greenberg 1987 [35]	30	19–150	44–73	4	Ciprofloxacin 57%	Prospective, randomized, open label
Dellamonica 1989 [34]	39	N/A	90–180	No mention	Pefloxacin 87% Ofloxacin 76% Ciprofloxacin 57%	Prospective
Mader 1990 [33]	26	29–60	28–64	No mention	10/12 (83%) IV patients 11/14 (79%) PO cipro patients	Randomized
Powers 1990 [32]	16	56	42	No mention	88%	Prospective
MacGregor 1990 [31]	18	35–364	140	None	61.6%	Retrospective
Gentry 1990 [86]	67	47	Cipro 56	No mention	24 (77%)	Randomized parallel group
Gentry 1991 [40]	33	28	56	No mention	74% Ofloxacin 86% IV therapy	Randomized
Eron 1992 [39]	53	45	45	None	74%	Open evaluation
Yamaguti 1993 [30]	17	None	28–254	None	76%	Prospective, open label
Gomis 1999 [38]	32	30–45	30–45	No mention	IV 70% PO 91%	Prospective, randomized, open label
Shih 2005 [91]	23	14	28	No mention	13/13 (100%)	Prospective
Esposito 2007 [92]	239	71.2	None	1 patient	107 (89.2%)	Retrospective analysis of OPAT in Italy
Cordero-Ampuero 2009 [28]	36	5	180	No mention	88%	Prospective
Estes 2010 [93]	20	42	270	No mention	90%	Retrospective
Conterno 2013 [7]	248	51	31	4 of 42 patients	All 8 trials reported cure	Systematic review
Babouee 2014 [26]	61	19	38	No mention	97%	Retrospective
Asseray 2016 [37]	230	N/A	91	3	At end of treatment: 40% After 1 year: 63%	Retrospective cohort
Laghmouche 2017 [25]	67	14	28	None	79.1%	Retrospective cohort
Fantoni 2019 [2]	50	<3 days	4–6 weeks	No mention	45 (90%)	Descriptive
Li HK et al. 2019 [15]	1054	6 weeks. Continued beyond 6 weeks for 76.7% of participants. Median duration in IV group: 78 days.		No difference in patients reporting at least one serious adverse event between IV and PO groups. Incidence of catheter complications was significantly higher in the IV group.	Oral antibiotics were non-inferior to IV antibiotics based on treatment failure rates (13.2% and 14.6%, respectively) at one year.	Randomized controlled trial
Frieler 2020 [20]	27	14	66–92	Line-associated complications, AKI	90%	Prospective cohort
Azamgarhi 2021 [49]	328	43–84	42–84	4	73.3% IV 85.7% PO	Pre/post cohort
Charalambous 2022 [82]	55	42	42–77	No mention	47%	Retrospective

## Data Availability

Not applicable.

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
