# Peer review of "A Review of the Clinical Utilization of Oral Antibacterial Therapy in the Treatment of Bone Infections in Adults"

_antibiotics, 2023, doi:10.3390/antibiotics13010004_

Round 1
Reviewer 1 Report
Comments and Suggestions for Authors
Dear Authors,
The theme chosen by you for this review is interesting, regarding the question if the oral antibacterial therapy may replace the need of the parenteral regimen required for the treatment of bone infections. You limited your literature search to years 1975-2023 and to the adult infections.
In my opinion, there are some aspects that I suggest to be corrected or added, in order to improve the scientific content of your manuscript:
1. you present the antibiotics used in different studies, but this presentation is a little superficial, especially that your paper wants to be a literature review. The antibiotics are frequently only enumerated. There are no regimens mentioned and the doses used are totally absent.
2. I suggest using mote tables and figures, in order to resume the information. There are too long paragraphs, which make reading difficult
3. the title of Table 1 has to be changed, because you say "...oral antibiotics...", but there is Fluconazole in the table, which, obviously, is not an oral antibiotic.
4. the names of the pathogens need to be written in Italic. Some of them are, some not
5. in lines 294-296, you talk about oxacillin, nafcillin and cefazolin, saying that "unfortunately, these agents are not available in oral formulation...". This is not true, there are oral forms of oxacillin.
6. detail what PJI in line 344 stands for....also for TKA in line 345
7. the recommendations presented in lines 636-647 need to be more organized
8. The references chosen are in agreement with the theme
9. there are some English mistakes that have to be corrected, such as the lack of "the" in many phrases or of "of"
Comments on the Quality of English Languagethere are some English mistakes that have to be corrected, such as the lack of "the" in many phrases or of "of"
Author Response
Thank you for the time you invested in the review of our manuscript. I will respond to each item:
- We have tried to improve the flow to avoid any semblance of being a 'little superficial'. We do believe the strength of this paper stems from the extensive literature reviewed, that lead to some clinically utilizable recommendations. Regarding antimicrobial regimens, they are presented in table 1, starting line 75. If there is anything I am misunderstanding from this question, please let me know. The table has doses and frequencies.
- There are two tables in this manuscript. Given the paper's descriptive nature, the authors felt those two tables encapsulated the objectives and findings we needed to convey. Specifically, Table 2 summarizes all studies in the literature that reviewed more than 15 patients, whether those designs were randomized, prospective, retrospective or systematic reviews.
- Thank you for pointing this out. Fluconazole is removed from table 1.
- Thank you also for pointing this out. All of the bacterial Genus sp. are now italicized.
- Very astute observation. In the US, oral oxacillin per se does not exist. But based on your observation that it existed elsewhere, we made that change per your recommendation.
- Corrected. Thanks for noticing them.
- We agree, the recommendations needed to be clarified and better organized, which we have done.
- Thanks for this comment
- Okay- made some changes as much as we could point this syntax issue out.
Reviewer 2 Report
Comments and Suggestions for Authors
In this review article, the authors conduct a thorough review of the management of chronic osteomyelitis, with a focus on exploring the viability of oral antibiotics as an alternative to the standard intravenous approach. The paper critically assesses existing literature since 1975, emphasizing Staphylococcus aureus infections and considering subpopulations like diabetic patients and those with implanted hardware. The advantages of oral antibiotics, including the avoidance of complications and costs associated with intravenous therapy, are underscored. The dual objectives of providing clinical recommendations and summarizing specific oral antimicrobial agents, considering their pharmacokinetic/pharmacodynamic properties and optimal duration of therapy, are clearly outlined. However, the authors acknowledge the limitations of existing evidence due to the scarcity of randomized controlled trials, highlighting the need for further research to establish more robust clinical guidelines. Overall, the manuscript demonstrates good writing and effective execution. However, some areas could be improved to enhance the quality and impact of the manuscript.
1. The introduction and conclusion can be more succinct.
2. Italicize the isolate name consistently throughout the manuscript, as seen in table no 1, line 216, 264, 265, 274, 277, 311, etc.
3. The author needs to individually discuss challenges and considerations in existing approaches, such as the complications of IV antimicrobial therapy, inconvenience, and the need for alternatives.
Author Response
Thank you for the time you invested reviewing our manuscript. We deeply appreciate the comments you provided, and herein we are responding to them:
- We have reorganized/restructured the introduction and conclusion such that they flow better, with less redundancy and an overall improved content.
- Thank you for trying to point them out-- we went over all of them again, and italicized what needed to be italicized.
- We believe these concerns have been mentioned in the paper on a couple occasions. Inasmuch as your input is astute in this situation, the authors feel that delving into the side effects/complications of IV vs. PO therapy is another large topic that does not totally fit within the objectives of this particular review. The manuscript is long already, and we aimed at avoiding additional sections. For that reason, the mention of complications and inconveniences of IV therapy was cursory and not exhaustive.
Round 2
Reviewer 1 Report
Comments and Suggestions for Authors
Dear Authors,
Thank you for considering my comments and suggestions and for making the changes in the manuscript.